# A Novel Ferroptosis Inhibitor UAMC-3203, a Potential Treatment for Corneal Epithelial Wound

**DOI:** 10.3390/pharmaceutics15010118

**Published:** 2022-12-29

**Authors:** Anusha Balla, Bao Tran, Annika Valtari, Philipp Steven, Camilla Scarpellini, Koen Augustyns, Arto Urtti, Kati-Sisko Vellonen, Marika Ruponen

**Affiliations:** 1School of Pharmacy, University of Eastern Finland, Yliopistonranta 1, 70211 Kuopio, Finland; 2Division of Dry-Eye and Ocular GVHD, Department of Ophthalmology, Faculty of Medicine and University Hospital Cologne, University of Cologne, 50923 Cologne, Germany; 3Laboratory of Medicinal Chemistry, Department of Pharmaceutical Sciences, Faculty of Pharmaceutical, Biomedical and Veterinary Sciences, Campus Drie Eiken, University of Antwerp, Universiteitsplein 1, B-2160 Antwerp, Belgium; 4Faculty of Pharmacy, University of Helsinki, 00014 Helsinki, Finland

**Keywords:** cornea, wound healing, UAMC-3203, ferroptosis

## Abstract

Corneal wound, associated with pain, impaired vision, and even blindness, is the most common ocular injury. In this study, we investigated the effect of a novel ferroptosis inhibitor, UAMC-3203 (10 nM–50 µM), in corneal epithelial wound healing in vitro in human corneal epithelial (HCE) cells and ex vivo using alkali-induced corneal wounded mice eye model. We evaluated in vivo acute tolerability of the compound by visual inspection, optical coherence tomography (OCT), and stereomicroscope imaging in rats after its application (100 µM drug solution in phosphate buffer pH 7.4) twice a day for 5 days. In addition, we studied the partitioning of UAMC-3203 in corneal epithelium and corneal stroma using excised porcine cornea. Our study demonstrated that UAMC-3203 had a positive corneal epithelial wound healing effect at the optimal concentration of 10 nM (IC_50_ value for ferroptosis) in vitro and at 10 µM in the ex vivo study. UAMC-3203 solution (100 µM) was well tolerated after topical administration with no signs of toxicity and inflammation in rats. Ex-vivo distribution study revealed significantly higher concentration (~12–38-fold) and partition coefficient (*K_p_*) (~52 times) in corneal epithelium than corneal stroma. The UAMC-3203 solution (100 µM) was stable for up to 30 days at 4 °C, 37 °C, and room temperature. Overall, UAMC-3203 provides a new prospect for safe and effective therapy for corneal wounds.

## 1. Introduction

The cornea is an avascular, transparent tissue covering the ocular surface. It is highly susceptible to damage by the external environment, such as allergic conjunctivitis due to allergens, injuries, and oxidative stress caused by chemical and thermal burns, thereby further leading to dry eye disease and optic nerve neuropathy [1,2,3]. Moreover, these damages may lead to a corneal wound, including impaired corneal nerves and nociceptors, that is characterized by impaired re-epithelialization of the corneal epithelium. The corneal wound is associated with intense pain, discomfort, and disability that can drastically affect visual function [4]. The corneal-wound-induced changes in corneal shape and structure may lead to corneal scarring resulting even in corneal blindness [5]. Several therapies, including conventional artificial tears and ointments for lubrication, prophylactic antibiotics, pressure patching, therapeutic contact lenses, amniotic membrane transplantation [6], topical growth factors [7,8], and human serum-derived and plasma-derived therapies [9,10], are used for the treatment of corneal wounds [11,12]. However, such therapies may provide only symptomatic relief and delayed healing [13,14,15]. Additionally, serum and plasma-derived therapeutics are cost intensive, time-consuming, and not accepted in several countries due to a lack of prospective randomized trials [16]. Therefore, safe, and effective drugs for corneal wound treatment are needed.

Ferroptosis is an iron-dependent form of regulated cell death that is mainly characterized by the accumulation of lipid reactive oxygen species (ROS) [17]. Ferroptosis has been studied in association with ischemia-reperfusion injury, kidney injury, cardiac diseases, and neurodegenerative diseases [18,19]. Its role in ocular diseases, such as corneal epithelial disease [20], dry eye [21], retinal pigment epithelial diseases [22], and glaucoma [23], has been recently investigated [24]. The association of ferroptosis with alkali corneal wound healing has also been studied. In the alkali burn-induced corneal injury mouse model, accumulation of ROS resulted in elevated expression of peroxide 4-hydroxynonenal (4-HNE), a lipid peroxidation by-product that can alter cell membrane permeability. Furthermore, decreased expression of glutathione peroxidase 4 (GPX4), an enzyme catalyzing the reduction in lipid peroxide, was seen [25]. Another study showed delayed wound healing in GPX4^+/−^ mice models after n-heptanol-induced corneal wounds, indicating the vital role of GPX4 [20]. Elevated ROS and lipid peroxidation with subsequent corneal ferroptosis has also been associated with exposure to heated tobacco products [26], aging [27], and ocular disease [28].

Ferrostatin-1 (Fer-1) is a specific ferroptosis inhibitor that protects corneal cells and has been shown to promote corneal wound healing [25,29]. Ferrostatin-1-loaded liposomes effectively alleviated ferroptosis; restored GPX4 levels; and reduced corneal edema, inflammation, and corneal neovascularization in the alkali-burn-wounded cornea [25]. Fer-1 also reduced cell death and improved cell viability in the human corneal epithelial cell line after exposure to cigarette smoke and heated tobacco products [26]. Poor water-solubility and hydrolytic instability of Fer-1 have limited its development in eyedrop products [30]. UAMC-3203 is an experimental ferroptosis inhibitor with higher in vitro potency (IC_50_ = 10 nM) than Fer-1 (IC_50_ = 33 nM) [30]. UAMC-3203 also has better metabolic stability than Fer-1 in human microsome (t_1/2, UAMC-3203_ = 20 h, t_1/2, Fer-1_ = 0.1 h) and plasma (recovery at 6 h: UAMC-3203 = 84%, Fer-1 = 47%) [30]. Compared to Fer-1, UAMC-3203 was more effective in a mouse model of (multi)organ dysfunction and death. As a preclinically safe and effective compound, UAMC-3203 is a potential drug candidate for ferroptosis-mediated diseases [31].

In the present study, we investigated the efficacy of UAMC-3203 as a corneal epithelial wound healing compound in mouse eyes with alkali-induced corneal wounds. We also studied the mechanism of wound healing effects (migration or proliferation) with in vitro scratch assays in human corneal epithelial (HCE) cells. We evaluated the acute tolerability in rats in vivo. Furthermore, the physicochemical properties of UAMC-3203 and its corneal distribution into excised porcine cornea were investigated.

## 2. Materials and Methods

### 2.1. Synthesis, Solubility, and Chemical Stability

Compound UAMC-3203 was synthetized in the laboratory of Medicinal Chemistry at the University of Antwerp, as reported by Devisscher L. et al. [30]. Solubility of UAMC-3203 was determined in 0.1 M citrate buffer (pH 5 and pH 6) and phosphate-buffered saline (PBS, Gibco, Life Technologies Limited, Paisley, UK) at pH 7.4. Excess of UAMC-3203 (10–20 mg) was added to 500 µL of buffer in glass vials that were kept at room temperature and mixed (150 rpm) for 72 h. The pH of the solutions was measured daily with calibrated pH meter (Orion Research Incorporated, Boston, MA, USA) and adjusted if needed (with 0.1 N sodium hydroxide or 0.1 N hydrochloric acid). After 72 h, samples were withdrawn and centrifuged at 13,000 rpm for 15 min. The supernatant was collected and analyzed by high-pressure liquid chromatography (HPLC) for UAMC-3203 concentration.

The chemical stability of 100 µM UAMC-3203 in PBS (pH 7.4) was investigated for 30 days as described previously [32]. Briefly, four batches with triplicates were stored at 4, 25, or 37 °C with light protection and at 25 °C without light protection. Separate solutions were used for pH-dependent stability studies. Samples were collected, and pH was measured at various times. The samples were stored at −20 °C until HPLC analysis.

The quantitation of UAMC-3203 was performed with Shimadzu Technologies’ reverse-phase HPLC system consisting of a degasser (G1379B), multi-sampler (G1367D), a binary pump (G1312B), a thermostat (G1316B), and a diode array detector (DAD) detector (G1315C). A Zorbax Eclipse XDB-C18 (4.6 × 50 mm, 4 μm; Agilent Technologies, Santa Clara, CA, USA) column was used for the analyses. The mobile phase was composed of 0.1% tetrafluoroacetic acid (Sigma-Aldrich, St. Louis, MO, USA) in Milli-Q-H_2_O and 0.1% tetrafluoroacetic acid in acetonitrile (LC-MS Chromasolv, Honeywell, Riedel-de Haen, Seelze, Germany) (65:35% *v*/*v*). Isocratic analysis was performed at a flow rate of 1 mL/min. The injection volume was 5 μL, and the detection wavelength was 240 nm.

### 2.2. Drug Distribution in Porcine Cornea Ex Vivo

Fresh porcine (a crossbreed between Matias and Yorkside) eyes were collected in the laboratory animal center at the University of Eastern Finland. The eyes were stored overnight in a keratinocyte serum-free medium (Gibco, Life Technologies Corporation, Grand Island, NY, USA) at +4 °C. The cornea was excised using an incision along the limbus, as described earlier [33].

The isolated corneas were rinsed with PBS and mounted in vertical Franz diffusion cells (PermeGear, Inc., Hellertown, PA, USA). Small magnetic stir bars were added to each receiver chamber, and 5 mL of 35 °C BSS Plus supplemented with 10 mM of 4-(2-hydroxyethyl)-1-piperazineethanesulfonic acid (HEPES; Lonza Bioscience, Walkersville, MD, USA) was added to the receiver compartment of each Frantz cell. The experiment was initiated by adding 50 µM UAMC-3203 in 10 mM HEPES-BSS Plus solution to the donor compartment. Samples of 500 µL from donor compartments and corneas were collected at the end of the experiments (at 5, 30, 60, and 120 min). The cornea samples were rinsed with PBS, the epithelial layer was scraped off from the corneas, and the remaining stroma–endothelium samples were cut to pieces. The experiments were carried out in triplicates. The samples were stored at −80 °C until liquid chromatography–tandem mass spectrometry (LC-MS/MS) analyses were performed.

The values of apparent partition coefficients (*K_p_*) were calculated using drug concentrations on the tissue samples (at 120 min) and in the donor compartment buffer at time zero (ng/mL).

### 2.3. LC-MS/MS Analysis

Quantitative analysis of UAMC-3203 was performed with the LC-MS/MS system that consisted of an Agilent 1290 series liquid chromatograph connected to an Agilent 6495 triple-quadruple mass spectrometer (Agilent Technologies, Inc., USA). The mobile phase consisted of 0.1% formic acid (Fisher chemical, Geel, Belgium), 1 mM ammonium formate (Honeywell, Fluka, Seelze, Germany) in Milli-Q-H2O (eluent A), and acetonitrile (LC-MS Chromasolv, Honeywell, Riedel-de Haen, Seelze, Germany) (eluent B). Gradient elution was used: 0.0 to 3.0 min: 25%→95% B, 3.0 to 3.5 min: 95% B, 3.5 to 3.6 min: 95%→25% B, 3.6 to 5.0 min 25% B. Solvent flow was 0.3 mL/min, and Poroshell 120 SB-C18 column (2.1 mm × 50 mm, 2.7 µm, Agilent Technologies, Inc., USA) was maintained at 60 °C. The injection volume was 2 µL. Nitrogen was used as a drying, nebulizer, and collision gas. The following conditions were used in the electrospray ion source: positive ion mode, sheath gas flow rate 11 L/min at 350 °C, drying gas flow rate 16 L/min at 200 °C, nebulizer gas pressure adjusted to 25 psi, capillary voltage 3500 V (ESI+), fragmentor voltage 380 V, dwell time 150 ms and cell accelerator voltage 5 V. The data were analyzed with Agilent Mass Hunter Quantitative Analyzed software (vB.09.00, build 9.0.647.0, Agilent Technologies, CA, USA). The precursor and fragment ions are presented in Appendix A.

### 2.4. In Vitro Scratch Assay

In vitro scratch assay was performed in confluent HCE monolayer as previously reported [34]. HCE cells were seeded on 24-well plates at 1 × 10^5^ cells/well incubated for 24 h. The monolayer was scratched with sterile 10 µL pipette tips perpendicular to the lines to create a consistent cell-free area. The cells were washed three times with pre-warmed PBS to remove detached cells. The cells were then exposed to UAMC-3203 at different concentrations (10 nM, 1 µM, 10 µM, 50 µM) in serum-free DMEM (Dulbecco’s Modified Eagle Medium): F12 media that was supplemented with 1% penicillin-streptomycin solution 1000 U/mL penicillin, 0.1 mg/mL streptomycin (Corning, Ref 30-002-CI, Mediatech Inc., Discovery Boulevard, Manassas, VA, USA). Serum-free medium was used as a negative control, and 1.5% fetal bovine serum (FBS 10270, Gibco by Life Technologies, Carlsbad, CA, USA) supplemented medium was used as a positive control. The well plates were then transferred to Cell-IQ (CM Technologies Limited, Tampere, Finland) and maintained in standard tissue culture conditions (37 °C, 5% CO_2_-atmosphere).

Images were taken immediately after scratch (less than 1 h) and at 6, 12, 24, 30, 42, 48, and 72 h and analyzed by Cell-IQ analyzer. Three independent experiments were performed, using two to three wells for each stimulating condition. Cell migration (*A*%) was calculated as follows:(1)$$A\%=\frac{A_{0}-A_{t}}{A_{0}}\times 100$$
where *A*_0_ and *A_t_* represent the wound areas at 0 h and each timepoint, respectively.

### 2.5. In Vitro Cytotoxicity

The cytotoxicity of UAMC-3203 was assessed by MTT assay. The HCE cells were seeded on a 96-well plate at a density of 100,000 cells/well in 200 µL of supplemented growth medium. The next day the cells were washed with PBS (pH 7.2). The cells were then exposed to UAMC-3203 at concentrations 10 nM, 1 µM, 10 µM, and 50 µM for 3 h. For control, cells not exposed to UAMC-3203 cultured in a serum-free medium were used. The cells were then washed twice with PBS and incubated 2 h with 100 µL of 10% of thiazolyl blue tetrazolium bromide (MTT, Sigma-Aldrich, St. Louis, MO, USA) in a serum-free medium. An amount of 100 µL of 20% (*w*/*v*) sodium dodecyl sulfate (SDS), N-dimethylformamide (DMF) lysis buffer was added to each well and incubated overnight. The following day, cell viability was evaluated by measuring absorbance at 570 nm with a Victor^2^ multilabel plate reader (PerkinElmer, Wallac, St. Paul, MN, USA). The cell viability % was calculated as below:(2)$$\%\;of\;cell\;viability=\frac{\left(Abs\;exposed\;cells-Abs\;blank\right)}{\left(Abs\;non-exposed\;cells-Abs\;blank\right)}\times 100$$
where Abs exposed cells = absorbance of cells exposed to UAMC-3203; Abs blank = absorbance of well plates without cells; and Abs non–exposed cells = Absorbance of cells exposed to serum-free medium (i.e., control).

### 2.6. Corneal Epithelial Wound Healing in Ex Vivo Mouse Eyes

For the study, C57BL/6 mice (Laboratory Animal Centre, the University of Eastern Finland) were euthanized, and the eyes were excised carefully with sterile forceps and micro-scissors to avoid any damage to the cornea. The eyes were transported in ice-cold Dulbecco’s modified Eagle’s Medium/Ham’s F12 (DMEM/F-12, Gibco, Life Technologies, Carlsbad, CA, USA). Then the eyes were adhered to a Petri dish with tissue adhesive (3M Vetbond, St. Paul, MN, USA). The corneal epithelial wound was produced by placing an alkali soaked (0.5% sodium hydroxide) Whitman filter paper of 2 mm diameter on the corneal surface for 2 min, as reported earlier [35]. After removal of the filter paper, the remaining epithelium on the alkali-exposed surface was carefully removed with a micro scalpel, and the eyes were rinsed with PBS. Pre-warmed UAMC-3203 (10 mL of 10 nM, 1µM, 10 µM, and 50 µM solutions) was added in serum-free media that was supplemented with 1% penicillin-streptomycin solution (1000 U/mL penicillin, 0.1 mg/mL streptomycin) (Corning, Ref 30-002-CI, Mediatech Inc., Discovery Boulevard, Manassas, VA, USA) to the Petri dishes so that the eyes were submerged. The serum-free medium without UAMC-3203 was used as the negative control, and the medium with 10% FBS was used as the positive control. The samples were cultivated under standard tissue culture conditions (37 °C, 5% CO_2_-atmosphere, and 95% relative humidity) for 48 h. In order to measure the wound area, images were taken with a Zeiss Axio Scope A1 microscope (Carl Zeiss Microscopy GmbH, Oberkochen, Germany) at different time points (initial (in less than 2 h after wound), 6, 24, 30, and 48 h). For imaging, the media was removed from the dish, and 0.1% fluorescein (2 µL) was added to the corneal surface, followed by PBS (10 µL) to rinse excess fluorescein. A clean tissue was then used to soak excess fluorescein and PBS, and the medium was added again to the dish after imaging. The medium was changed after 24 h. The wound area was quantified with Fiji software [36].

### 2.7. In Vivo Acute Tolerability Study in Rats

Three Lister-hooded rats (Envigo Laboratories, Melderslo, Limburg, The Netherlands) were used in the study. The animals were maintained under standard laboratory conditions of 12 h dark-light cycles with food and water available ad libitum. Animal studies demand set by EU directive 2010/63/EU, and all animal experiments were approved by the national Project Authorization Board (ESAVI/27769/2020).

The in vivo tolerability of 100 µM UAMC-3203 solution in PBS was determined after topical application by visual inspection, optical coherence tomography (OCT) (Phoenix MICRON™ MICRON IV/OCT, Pleasanton, CA, USA), and microscope imaging (Leica Stereomicroscope with fluorescence Immuno Diagnostic Ltd., LMS, Espoo, Finland). During the study, 10 µL of PBS (pH 7.4) was used as control, and 10 µL of UAMC-3203 solution was applied to the lower fornix of left and right eyes, respectively, twice a day (8 a.m. and 3 p.m.) for five consecutive days. The animals were allowed to move their head freely during the visual inspection while, before OCT and stereomicroscope imaging, the rats were anesthetized with a medetomidine (0.4 mg kg^−1^) and ketamine (60 mg kg^−1^) mixture. Baseline measurements for OCT and microscope imaging were taken two days prior to the study. Visual inspection was performed after each treatment, while OCT and stereomicroscope imaging was performed on the second and fifth days. The follow-up tests with OCT and stereomicroscope imaging were performed three days after the completion of the study.

Visual inspection—The eyes were checked visually for any symptoms of redness and eyelid swelling before every treatment. Then, after the application of PBS and UAMC-3203 solution, the number of times the rats blinked their eyes and moved their head within one minute were recorded.

OCT—The cornea was observed for any changes in its thickness and any signs of corneal edema during and after the treatment.

Stereomicroscope imaging—The cornea and sclera were evaluated for any signs of inflammation or neovascularization during and after the treatment.

## 3. Results

### 3.1. Solubility and Chemical Stability Studies

The water solubility of UAMC-3203 was about 3.5 times higher at pH 6.0 (127.9 ± 16.1 µM) and 7.4 (127.3 ± 17.3 µM) than at pH 5.0 (36.7 ± 5.7 µM). UAMC-3203 shows relatively high stability for 30 days in PBS pH 7.4 at various temperatures and light exposure conditions (about 90% of initial concentration at day 30). During the study period of 30 days, the pH of the UAMC 3203 solution slightly increased (from 7.4 to 7.5–7.6) in all conditions (Appendix A).

### 3.2. Drug Distribution in Porcine Cornea Ex Vivo

We studied the distribution of the UAMC-3203 (50 µM) in the cornea by measuring the concentration in corneal epithelial cells and in stroma–endothelium after compound exposure to the epithelial side of the cornea. The drug is distributed rapidly to the cornea, and the levels in the epithelium were 1–2 orders of magnitude higher than in the stroma–endothelium (Figure 1). The *K_p_* value of epithelium/donor (1.95 ± 0.45) was ≈52 times higher than the *K_p_* value of stroma/epithelium (0.04 ± 0.02).

### 3.3. Scratch Wound-Healing Assay In Vitro

In vitro scratch assay was used to assess the effect of UAMC-3203 in cell migration. The cell migration % was calculated by measuring cell coverage in the scratch (cell-free area) at defined time points. The results showed higher scratch closure in the presence of UAMC-3203 at concentrations 10 nM and 1 µM than at higher concentrations (10 µM and 50 µM) (Figure 2). At 72 h, the HCE cell migration (%) was significantly higher in the presence of UAMC-3203 at 10 nM (84.9 ± 12.5%) (*p* < 0.001) and 1 µM (76.6 ± 15.2%) (*p* = 0.003) compared to the negative control (55.6 ± 17.6%). Whereas at 10 µM concentration, there was no notable difference at any time point. The highest drug concentration (50 µM) did not cause any changes in the scratch area.

### 3.4. In Vitro Cytotoxicity Evaluation

We evaluated the effect of UAMC-3203 on HCE cell viability using an MTT assay. UAMC-3203 exposure of 3 h showed concentration-dependent cytotoxicity, as shown in Figure 3. The HCE cell viability was significantly reduced to 75 ± 6.7% and 39.2 ± 5.6% at 10 µM and 50 µM drug concentrations, respectively. Moreover, lower concentrations of 10 nM and 1 µM did not cause any toxicity.

### 3.5. Corneal Epithelial Wound Healing in Ex Vivo Mouse Eye Model

The corneal epithelial wound-healing effect of UAMC-3203 was studied using alkali burn corneal wounds in ex vivo mouse eyes. Compared to the initial wound area (0–2 h after wound), a significant decrease in wound area was seen during 48 h with the positive control (2.76 ± 0.62 to 0.12 ± 0.26 mm^2^), UAMC-3203 at 10 nM (2.59 ± 0.27 to 0.22 ± 0.16 mm^2^), 1 µM (2.46 ± 0.42 to 0.35 ± 0.3 mm^2^), and 10 µM (2.44 ± 0.52 to 0.10 ± 0.21 mm^2^) and negative control (2.72 ± 0.44 to 0.82 ± 0.54 mm^2^) (Figure 4).

Moreover, the optimal concentration for wound healing was 10 µM (significant difference in wound area at 24–48 h as compared to the negative control (*p* ≤ 0.005)). With positive control, a significant difference compared to negative control was obtained only at 30 h (*p* = 0.01), and it showed an equal effect with 10 µM UAMC-3203 at 48 h. Compared to the negative control, UAMC-3203 at 10 nM had a significantly lower wound area at 48 h (*p* = 0.03), whereas at 1 µM, there was no notable difference at any time point. The highest drug concentration (50 µM) did not cause any changes in the corneal epithelial wound area.

### 3.6. In Vivo Acute Tolerability Study in Rats

We studied the ocular safety of UAMC-3203 solution (in PBS) at a concentration of 100 µM. The rats were treated with topical administration (10 µL) of UAMC-3203) or PBS (used as control) twice a day for five consecutive days. The eyes were observed for clinical signs of toxicity and inflammation by visual inspection, OCT, and stereomicroscope imaging on day 2, day 5, and post-treatment on day 8, as shown in Figure 5A.

During the study, the eyes of both groups were free from any signs of irritation, such as redness and eyelid swelling. No comparable difference in the number of blinks and headshakes was observed between the control and treated groups (Appendix A). No significant change in the thickness of the corneal thickness was obtained between PBS and UAMC-3203 (Figure 5B). Similarly, no signs of corneal opacity, neovascularization, and inflammation, as redness was observed in the cornea and sclera of treated rats (Appendix A).

## 4. Discussion

Previously, the involvement of lipid-peroxidation-dependent ferroptosis was shown in alkali burn-induced corneal injury [25]. Moreover, other studies have shown links between corneal wounds and ferroptosis [21,25]. Therefore, we investigated UAMC-3203, a new ferroptosis inhibitor, as a potential treatment for corneal epithelial wounds.

In order to gain more insights into corneal wound healing and evaluate the potential therapeutic or toxic effects of compounds, various ex vivo animal models were developed with wounds induced by either chemical burn [35] or physical injury [37,38,39,40]. Ex vivo models allow investigations of corneal wound healing using experiments with controlled drug concentrations and exposure times. These models are in line with the 3R principle of laboratory animal use, providing valuable information on epithelial repair at a lower number of animals, thereby augmenting the design of in vivo experiments in drug development [41]. The models may also involve submerged tissue cultures [35,39], tissue cultures at air–liquid interface [37,38,40], free-floating and agar-mounted corneal disks [39], or whole eyes fixed to the culture plates [35]. Our model is modified from a combined in vivo/in vitro model that allowed initial partial healing for 6 h in vivo before excision and adhesion of bulbi in well-plates afterward [35]. In our model, we adhered the whole eye to the dish and induced the wound by alkali burn, as a previous study [25] has shown that such a severe wound resulted in ferroptosis in vivo. It is a simple model that enables the evaluation of specific effects of compounds for a corneal epithelial wound.

In our study, we saw fast and effective wound healing (>85%) at 10 nM–10 µM of UAMC-3203. Interestingly, we observed a positive wound-healing effect at the in vitro IC_50_ value for ferroptosis inhibition (10 nM) of UAMC-3203 [30]. Healing was also shown in in vitro scratch model at 10 nM and 1 µM of UAMC-3203, while the effects were less beneficial at higher concentrations. At 50 µM, UAMC-3203 was probably toxic, and no wound healing was seen during 48 h. These results were consistent with our in vitro scratch assay and MTT assay. Moreover, when tested in rats in vivo as a twice-daily topical application for 5 days, UAMC-3203 at 100 µM concentration did not result in any signs of toxicity (visual inspection, OCT, stereomicroscope imaging). High-resolution in vivo imaging with OCT [42,43] did not show differences in corneal thickness between the control group and UAMC-3203 treated animals. The stereomicroscope images did not show any corneal opacity or neovascularization, which is the most common nonspecific response to corneal wounds, inflammation, and hypoxia [44]. The differences in toxicity of UAMC-3203 may be attributed to different experimental setups, as in the ex vivo model, the eyes are submerged in the solution for 48 h, whereas the instilled eyedrops are eliminated from the ocular surface within 5 min [45], resulting in a rapid decrease in drug concentration on the cornea [46,47]. Based on these results, the corneal epithelial wound healing effects of the compound could be investigated in follow-up in vivo studies along with its role in corneal scarring.

Corneal wound healing involves four continuous yet distinct processes. The initial latent phase (4–6 h) without visible changes in wound size (Figure 4A) is characterized by an increased intracellular synthesis of proteins. During cell migration, the movement of cells from the limbus toward the central cornea is observed. Then, in the cell proliferation step, mitosis and differentiation of cells occur, followed by the final step involving cell attachment to the basal cell layers. As shown in our in vitro study, UAMC-3203 seems to accelerate corneal epithelial wound healing via cell migration.

In our ex vivo drug distribution study in the porcine cornea, we showed a preferential distribution of UAMC-3203 to the corneal epithelium. At 5 min, we observed rapid distribution of UAMC-3203 into the cornea at levels that were three orders of magnitude higher (≈20 µM) than the IC_50_ value of UAMC-3203 (10 nM). Lipophilic epithelium concentrations of the drug were ≈28 times higher than in the hydrophilic stroma at 120 min. Such difference in the distribution between corneal layers may be explained by the lipophilic (Log D_7.4_, 0.95) nature of the compound. Furthermore, a 52-fold higher *K_p_* epithelium/donor value was seen compared to *K_p_* stroma/epithelium, supporting the preferential distribution of UAMC-3203 corneal epithelium.

## 5. Conclusions

Corneal epithelial wound healing effects of new ferroptosis inhibitor UAMC-3203 were investigated. In this study, we demonstrated that UAMC-3203 is involved in the wound healing response in vitro and ex vivo. It can be an effective drug for corneal epithelial wound healing. Mechanisms of wound healing effects and long-term efficacy of UAMC-3203 should be further explored to determine the potential for further translation.

## Figures and Tables

**Figure 1 pharmaceutics-15-00118-f001:**
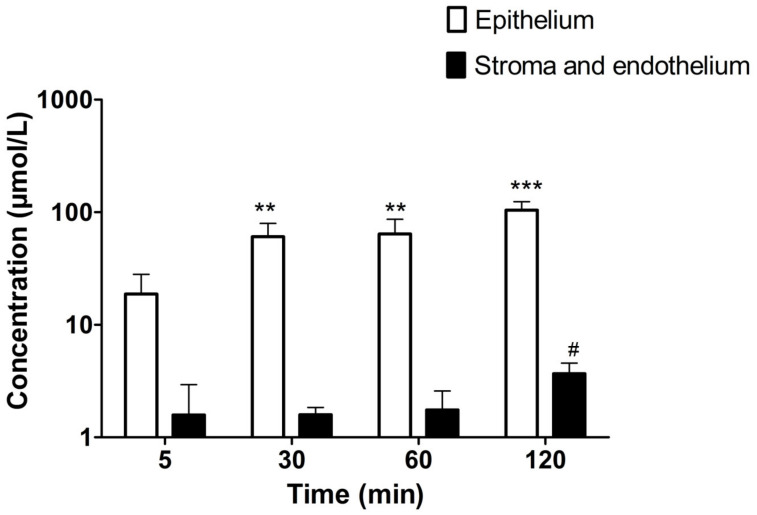
The concentrations of UAMC-3203 (50 µM) in corneal epithelium and stroma with endothelium at various time points. Statistical analysis was performed by *t*-test: ** *p* < 0.01 and *** *p* < 0.001 for concentration µmol/L epithelium (compared to that at 5 min), and # *p* < 0.05 for concentration µmol/L and endothelium (compared to that at 5 min). The results are expressed as mean ± standard deviation, *n* = 5–6.

**Figure 2 pharmaceutics-15-00118-f002:**
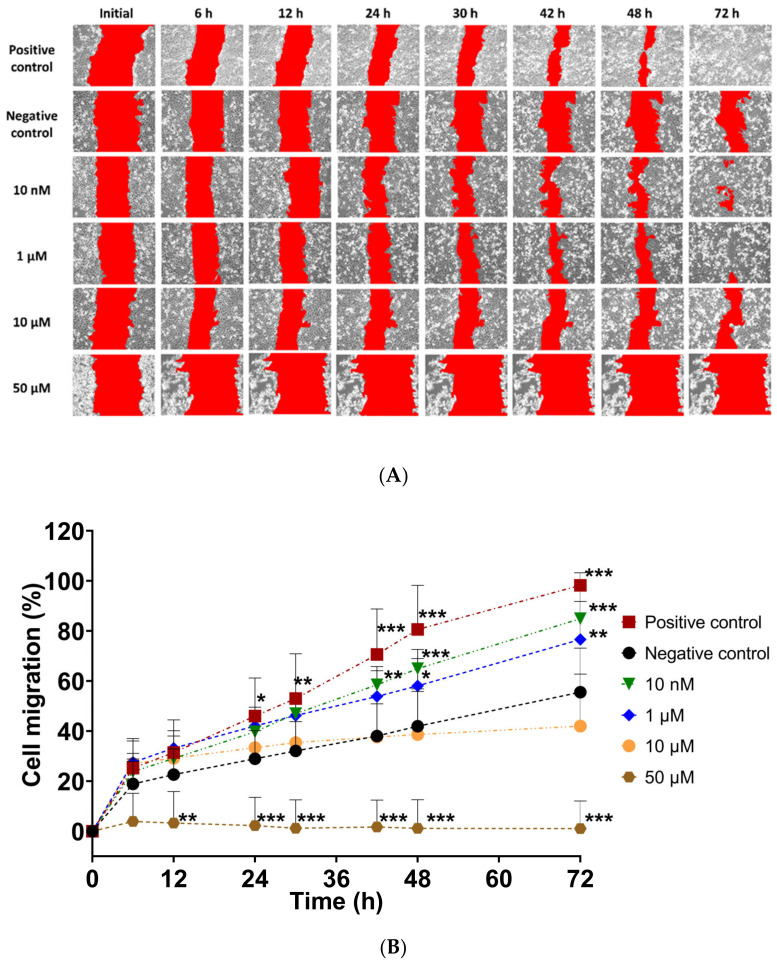
(**A**) Representative images following in vitro scratch in confluent HCE monolayer cells, and (**B**) measurement of the cell migration (A%) at the indicated time for cells treated with DMEM medium supplemented with 1.5% FBS (positive control), serum-free medium (negative control), and UAMC-3203 at concentration 10 nM, 1 µM, 10 µM, and 50 µM. Statistical analysis was performed by two-way ANOVA compared to negative control with Bonferroni *t*-test: * *p* < 0.05, ** *p* < 0.01, and *** *p* < 0.001 (*n* = 8).

**Figure 3 pharmaceutics-15-00118-f003:**
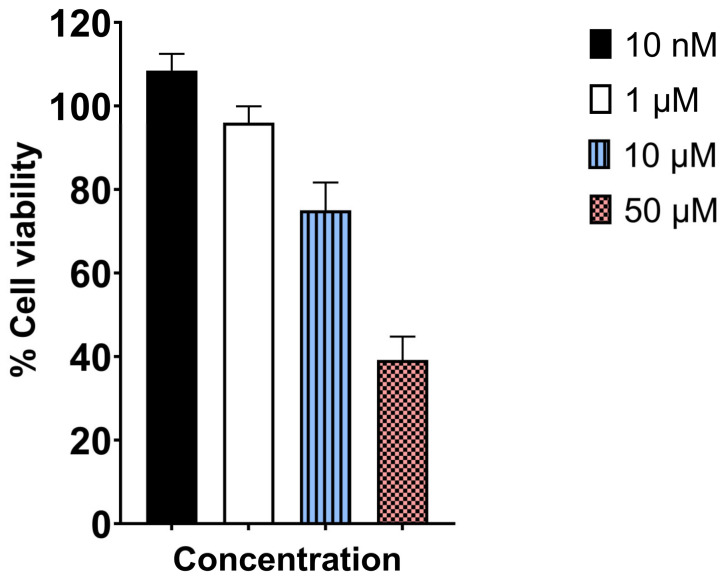
MTT assay determined cytotoxicity profile of UAMC-3203 after 3 h incubation with HCE cells at varying concentrations (*n* = 8).

**Figure 4 pharmaceutics-15-00118-f004:**
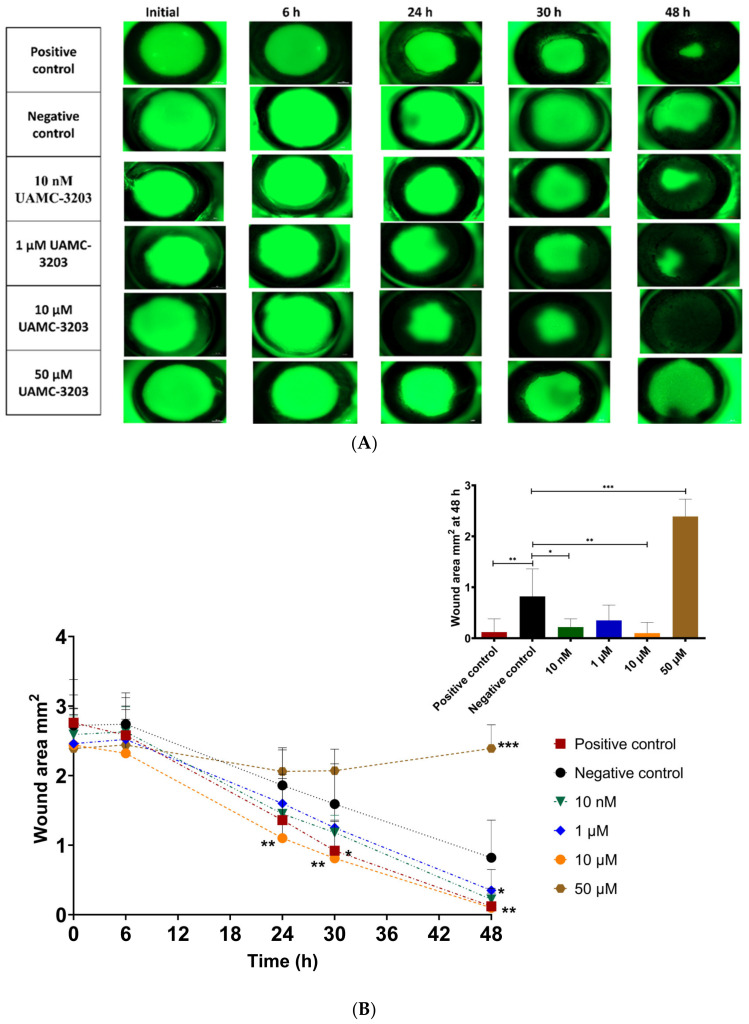
(**A**) Representative corneal photographs following fluorescein staining, and (**B**) measurement of the corneal epithelial wound areas (mm^2^) of alkali-wounded corneas treated with DMEM medium supplemented with FBS (positive control), serum-free medium (negative control), and UAMC-3203 at concentration 10 nM, 1 µM, 10 µM, and 50 µM at five time points during wound healing (decrease in wound area). Statistical analysis was performed by two-way ANOVA compared to negative control with Bonferroni *t*-test: * *p* < 0.05, ** *p* < 0.01, and *** *p* < 0.001 (*n* = 8–9).

**Figure 5 pharmaceutics-15-00118-f005:**
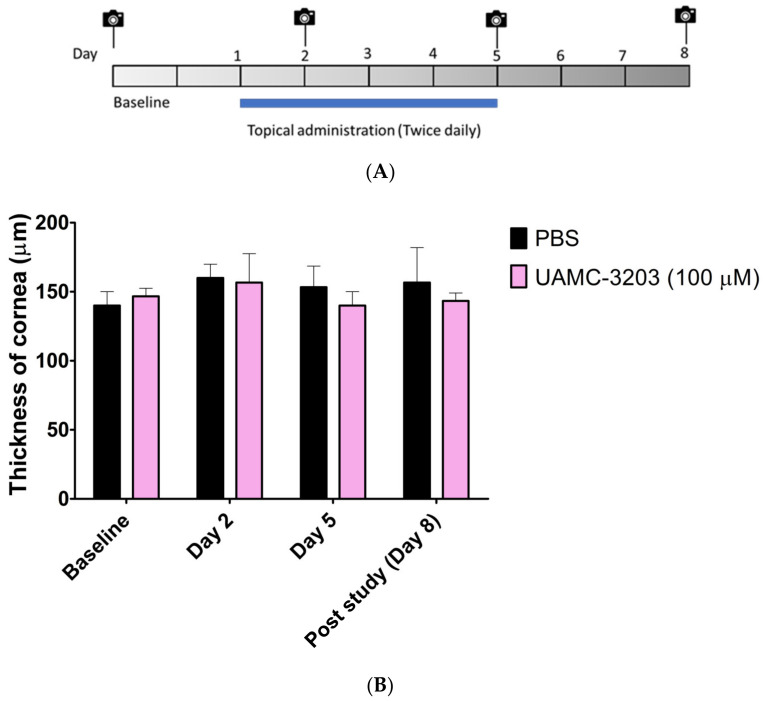
(**A**) Schematic diagram of administration, OCT, and stereomicroscope imaging in mice; (**B**) thickness of cornea (µm) in the control (PBS) and treated (UAMC-3203, 100 µM) groups. Corneal thickness in rats was determined with optical coherence tomography. The results are expressed as mean ± SD, *n* = 3.

## Data Availability

The data presented in this study are available on request from the corresponding author.

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
