# Peer review of "A Novel Ferroptosis Inhibitor UAMC-3203, a Potential Treatment for Corneal Epithelial Wound"

_pharmaceutics, 2022, doi:10.3390/pharmaceutics15010118_

Round 1
Reviewer 1 Report
This paper evaluated the effects of UAMC-3203 on corneal epithelial cell migration in vitro and in vivo. The study design is fairly straightforward and described well. Suggestions are made to improve the quality of the study.
-Please clarify in the text why the authors performed an ex vivo wounding study on the mice, instead of in vivo which would be more physiologically relevant.
-Generally, therapeutics targeted to promote corneal wound healing following injury focus on corneal scarring, but there is no mention of this in the intro or implemented in the study design.
Author Response
Thank you for the comments and suggestions to the manuscript. The point-by-point response can be found in the attached PDF. The changes in the manuscript are highlighted in yellow.

Reviewer 2 Report
The authors report on the development of a formulation of ferroptosis inhibitor for treatment for corneal wound, which was tested in an animal model. Although the topic is interesting, it does not have enough novelty. For instance, Wang K has reported Ferrostatin-1-loaded liposome for treatment of corneal alkali burn via targeting ferroptosis (Bioeng Transl Med. 2021 PMID: 35600640; PMCID: PMC9115688).
Furthermore, there are also several issue that must be addressed:
1. Please elaborate why you choose alkali burn model and not using corneal epithelial defect model for this study.
2. I suggest that adding experiment to evaluate the effect of UAMC-3203 on corneal Corneal nerve.
3. Why do you choose alkali burn model and not using corneal epithelial defect model for this study?
4. I recommend use corneal alkali burn experiment in vivo instead of corneal wound healing in ex vivo mouse eyes.
5. It is better to supplement the positive and negative control to judge the effect of UAMC-3203 on HCE cell viability using MTT assay.
Figures
Figure 2. The initial picture of the Scratch wound seems to be heavier in the positive/negative/drug concentration (50 µM) than other groups, please give an explanation.
Author Response

(The authors gave the same response as above.)

Round 2
Reviewer 2 Report
This paper addressed an important and interesting study,the work has been improved on the basis of the referees' suggestions.